# Psychometric validation of the Thai version of the 12-item Zarit Burden Interview among informal caregivers in a palliative care setting in the south of Thailand

Ueamporn Summart[1], Thanaporn Kaewbubpha[2], Ornicha Saengnil[2], Prasertsri Kongkaur[2], Suparuek Mahorasop[2], Suchittra Saengkhum[3], Yuwadee Wittayapun[2,4]*

**1** Faculty of Nursing, Roi Et Rajabhat University, Roi Et, Thailand, **2** School of Allied Health Sciences, Walailak University, Nakhon Si Thammarat, Thailand, **3** Chaiya Hospital, Surat Thani, Thailand, **4** Movement Science and Exercise Research Center-Walailak University (MoveSE-WU), Nakhon Si Thammarat, Thailand

* Yuwadee.wi@wu.ac.th

## Abstract

The Thai version of the 12-item Zarit Burden Interview (ZBI) is valid for measuring caregiver burden in psychiatric caregivers. Nevertheless, it has not been verified in a larger palliative care (PC) context. The aim of this study was to validate the 12-item ZBI in PC among caregivers of palliative care patients. A methodological study examining the validity of the 12-item ZBI was carried out with informal caregivers of palliative care patients of two district health service networks. The data were gathered using self-administered questionaries, and then all participants were randomly split into two sub-samples for exploratory factor analysis (EFA) (n = 150) and confirmatory factor analysis (CFA) (n = 155). After utilizing statistical approaches for reducing items, EFA was used with group 1 to analyze the factor structure of the 12-item ZBI. Finally, CFA was employed with group 2 to confirm the amended structure indicated by the EFA and to evaluate the construct validity of the 12-item ZBI. A total of 305 palliative caregivers were enrolled. The principal component analysis of the 12 items yielded a loading based on a two-factor model of personal strain and role strain accounting for 61.4% of the variance. Cronbach's alpha (0.83) and item-total correlations (rho = 0.38–0.70) showed that the 12-item Zarit had acceptable reliability. For convergent validity, the average variance extracted (AVE) values revealed that all 12-item ZBI subscales had a convergence effect, with AVEs ranging from 0.50–0.53. Additionally, this tool had a significant positive correlation with depressive symptoms (r = 0.48), anxiety (r = 0.38), and stress (r = 0.56). The 12-item ZBI is a brief, precise, and valid instrument for assessing burden among Thai palliative caregivers. We discovered high evidence of reliability in this sample, along with convergent and construct validity. Likewise, EFA revealed that the 12-item ZBI was a two-dimensional

**Data availability statement:** All relevant data are within the manuscript and its Supporting Information files.

**Funding:** The author(s) received no specific funding for this work.

**Competing interests:** The authors have declared that no competing interests exist.

scale. Thus, health care practitioners may utilize the ZBI in research and clinical settings to determine burden in palliative caregivers.

## Introduction

The World Health Organization (WHO) redefined palliative care with an emphasis on enhancing the quality of life (QOL) of patients with life-threatening illnesses via early detection and proactive treatment of medical, psychological, and spiritual problems [1]. According to this description, palliative care (PC) meets the requirements of patients and provides a support framework to help families cope with the illness and their eventual loss [2]. Because of the increased number of people with advanced illnesses who prefer to die at home and a corresponding decline in the accessibility of professional home care, growing demands are placed on family caregivers to care for patients with life-threatening illnesses at home [3].

There are 2 types of caregivers formal and informal. The formal caregiver is the individual who is compensated for the service. The informal caregiver is typically a family member, friend, or neighbor. Although this type of caregiver is neither paid nor trained to conduct the function, they have a great degree of dedication as evidenced by their affective relationship [4]. In the case of palliative patients, the type of care in healthcare is usually informal, with the family member being the primary caregiver. This situation has impacted numerous caregivers of individuals with advanced or terminal illnesses, leading to a significant burden [5].

In general, three main sickness trajectories to mortality have been described: abrupt loss of functionality (e.g., aggressive cancer), variable deterioration (e.g., organ failure), and progressive decline (e.g., frailty or dementia) [6]. As a result, caregivers for patients receiving palliative care have reported an enormous burden related to their caring tasks [7], and over 75% of them have also reported sleep issues, weariness, pain, and depression [3]. Likewise, the caregiving burden has been demonstrated to relate to caregivers' depression and low QOL [8]. There is an increasing number of intervention programs that aim to enhance caregiver outcomes, such as reducing the caregiver burden, improving caregiver coping skills, or improving QOL [9]. However, it is impossible to assess the impact of interventions in the absence of validated caregiver outcome measures [10]. Hence, to identify caregivers who require intervention, healthcare personnel must quantify their burden using a valid and reliable instrument.

The Zarit Burden Interview (ZBI), a self-report questionnaire, is one of a standardized and validated tool, which has been used widely for the assessment of caregiver burden [11–17], assessing the impact of caregiving on caregivers in terms of physical, mental, social, and economic factors [18]. These self-report questionnaires provide various advantages in clinical settings, including minimal implementation costs and shorter administration times. The original version of the ZBI consisted of 22 items; however, shorter versions, such as the 12-item ZBI, have been developed to provide a brief and accurate assessment of caregiver burden [12]. This tool has been translated into different languages and evaluated for caregivers of a variety

of patients with physiological and psychological disorders, including dementia, heart failure, cancer, and other conditions requiring palliative care [10,15,19–21]. The ZBI is a unidimensional scale; however, other researchers contend that caregiver stress is multidimensional and cannot be effectively measured with an aggregate score [18]. The most typically reported two-factor structure of the 12-item ZBI is that of personal and role strain [16], but the factor structures of this tool are conflicting, ranging from one- to three-factor structures [11–17]. Furthermore, based on a literature analysis, we discovered diverse factor structures of ZBI in different studies, which appear to be related to the cultural backgrounds of the samples and the statistical methodologies used [15,22].

In one Thai study, the psychometric properties of the 12-item ZBI were tested in psychiatric caregivers. This study demonstrated a three-dimensional structure with excellent internal consistency [15]. However, since the sample of this study consisted of caregivers of psychiatric cases, it may have only exhibited the ability of the ZBI to detect mental health problems in a specific group, namely, palliative caregivers caring for patients with physiological disorders. Another Thai study examined the psychometric properties of both the full-length ZBI-22 and the ZBI-12 among a sample of Alzheimer's disease caregivers and reported that only the ZBI-22 demonstrated reliability and a unidimensional scale among this population [23]. Since caregivers have differing levels of involvement when caring for patients with different conditions, the palliative population differs from that of psychiatric disorders due to the rapid progression of diseases and the high caregiver burden at the end of life [10].

To date, no research has been undertaken among Thai palliative caregivers on the factor structures and convergent validity of the 12-item ZBI. Applying the original subscale scoring to both physiological and psychological illnesses without understanding the instrument's psychometric features may result in erroneous conclusions. Therefore, the objective of this study was to validate the psychometric properties of the 12-item ZBI in Thai caregivers of palliative patients.

## Materials and methods

### Study design and population

This methodological study focused on the validity of the 12-item ZBI, utilizing data from the "palliative care outcomes and factors associated with the quality of life of caregivers of palliative care patients" study[24]. This study was conducted in two district health service networks, which included two district hospitals and 24 sub-district health promotion hospitals. Data collection was conducted from December 26, 2023, to March 5, 2024, using self-assessment questionnaires. The informal caregivers of palliative patients served by the hospital support team and the home care team were recruited consecutively. The inclusion criteria for this study were the primary caregivers who included friends, family members, neighbors, and individuals involved in PC, through patients receiving treatment in medical institutions or at home through two district health service networks. They were male or female Thai nationals, aged 18 or above, who had provided care for a minimum of 3 days per week for at least 3 months without payment [25]. The exclusion criteria were caregivers with poor general conditions or having mental disorders such as schizophrenia, psychosis, or dementia.

### Study instruments

The first section of the questionnaire was composed of caregiver demographic data, including gender, age, educational level, marital status, and relationship with the palliative patient, and the diagnosis of the patient.

The Thai version of the 12-item ZBI (ZBI-12) has been validated in family caregiver of psychiatric patients with an excellent internal consistency [12]. This tool consists of 12 items measured on a 5-point Likert scale ranging from 0 (never) to 4 (always), with a total score range of 0–48. Higher total scores indicate a higher burden, with a score of 20 or more deemed a significant burden [12]. The abbreviated Thai version of the burden scale assesses caregiver burden as a global score and is comparable to the 22-item version [23].

The Thai version of DASS-21 is a combination of three self-report scales used to evaluate depression, anxiety, and stress. It has 21 items separated into three subscales (items 3, 5, 10, 13, 16,17, and 21 for the 'depression' subscale;

items 2, 4, 9, 15, 19, and 20 for the 'anxiety' subscale; and items 1, 6, 8, 11, 12, 14, and 18 for the stress subscale). The DASS-21 items can be scored on a four-point scale. This scale produces scores for each of the three subscales, with higher scores indicating greater symptom intensity. According to the DASS manual, subscale scores are categorized as normal, mild, moderate, severe, or extremely severe. Moreover, a recent study in Thailand has validated this tool and reported good psychometric properties [26]. The Cronbach's alpha coefficient for the Thai version of the DASS-21 was 0.92.

## Sample size calculation

The sample size was calculated using the "sample size = (number of items) x (number of participants)" formula, which is widely used in survey development research. We calculated the required sample size using one item per 10 participants [27]. Based on the number of items in the ZBI (n = 12), a minimum acceptable sample size of 120 respondents was determined. Thus, 305 palliative caregivers were included in this cross-sectional study. To minimize model overfitting, the exploratory factor analysis (EFA) and confirmatory factor analysis (CFA) were performed on individuals randomly divided into two groups (group 1, n = 150, and group 2, n = 155).

## Statistical analysis

All statistical analyses in this study were conducted using IBM SPSS and AMOS version 20. The participants' demographic data were described using descriptive statistics, which included the means and standard deviations for continuous variables and numbers and percentages for categorical data.

The floor and ceiling effects of the items were examined, considering the percentages of respondents who chose the lowest and highest answer alternatives, respectively. In this regard, items with percentages of 15% or less were considered free of these effects. Furthermore, the discriminating ability of the items was tested by using corrected item-rest polyserial correlation, with acceptable indices greater than 0.20 [28].

To determine the number of components in the EFA of the 12-item ZBI measurement model, parallel analysis (principal component analysis) was used with sample group 1. The structure of the factors was then investigated using principal axis factoring and varimax rotation. Factor loadings of less than 0.5 were muted, while item cross-loadings of more than 0.2 were deleted one at a time. Furthermore, factor loadings were used to calculate the average variance extracted (AVE) and composite reliability (CR). Following the determination of the principal axis factoring results, a CFA was applied to the remaining participants. This measurement model was fitted using the covariance matrix and the unweighted least squares method, and an acceptable model was conventionally indicated by the fit indices, including the cumulative fit index (CFI), adjusted goodness of fit index (AGFI), root mean square error of approximation (RMSEA), and Tucker-Lewis index [29,30]. Similarly, a root mean square error of approximation (RMSEA) with a p-value less than 0.08 was assumed to indicate a good model fit; hence, it was reported and utilized in this analysis by convention. In addition to the CFA, the Kaiser-Meyer-Olkin (KMO) measure of sampling adequacy and Bartlett's test of sphericity were developed to provide evidence of construct validity [31].

Reliability was assessed using CR and Cronbach's alpha. CR is considered appropriate when the values for all three subscales exceed 0.6 [32]. Internal consistency reliability was assessed using Cronbach's alpha, and Cronbach's alpha values greater than 0.7 for all subscales were judged satisfactory [33]. Item-total and item-item correlations were used to assess item homogeneity and internal consistency; correlations greater than 0.20 were deemed satisfactory [33]. Additionally, item-item correlations were considered acceptable if they ranged between 0.30 and 0.70.

We used the AVE to explore the convergent validity of the 12-item ZBI. To be considered for convergent validity, the AVE must be equal to or greater than 0.50, indicating that the construct's variance determines more than half of its variability [30]. Concurrent validity was measured using Pearson correlation with expected significant positive correlation with DASS-21, assuming that high correlation implies high convergent validity and suggests that the two scales assess similar

concepts [34]. By following Cohen's criteria, the correlation between the variables was classified as small, medium, or large, with correlation coefficients greater than 0.10, 0.30, and 0.50, respectively [35].

Furthermore, the Fornell and Larcker criterion, which consists of comparing the square root of the AVE and the correlations with the other variables, was used; the former must be greater than the latter to determine that there is discriminant evidence [36].

## Ethical considerations

The study followed the ethical standards and principles outlined in the Declaration of Helsinki. Ethical clearance for the study protocol was granted by the Walailak University Institutional Review Board (Reference No. WUEC-23-344-01). Each of the study participants provided informed written consent for this investigation. Furthermore, each participant was assured of the privacy of his personal information.

## Results

### General information of the participants

The details of the participant characteristics are described in **Table 1**. Most participants (79.4%) were female, ranging in age from 30 to 86 years old ($\bar{x}$ = 54.18; S.D. = 13.52 years) About two-thirds of the participants (71.4%) had completed secondary education and were married (68.2%). About half of the participants were patients' children (48.8%). The most common diagnosis among patients was a neurological illness (33.1%).

### Item analysis

Floor effects were observed for all items of the 12-item ZBI. In terms of discriminative ability, all of the items had polyserial item-rest correlation coefficients greater than 0.20 (**Table 2**).

### Exploratory factor analysis

After randomly assigning participants to two groups (150 for EFA and 155 for CFA), parallel analysis was performed, and a suitable two-factor structure was established. The initial analysis of the group 1 sample resulted in two components with eigenvalues greater than 1. The factorial structure of the 12-item ZBI and the underlying dimensions that comprised its 12 items were determined. The initial analysis indicated a two-factor structure that accounted for 61.74% of the original data's variation. The KMO value was 0.87, and the Bartlett sphericity test result was significant ($\chi 2$ = 2009.490; p < 0.001), indicating that the data were sufficient for the factor analysis (**Table 3**).

### Confirmatory factor analysis

An unweighted least squares CFA was used to fit the 12-item ZBI measurement model, which had 12 items divided into two components: personal strain (10 items) and role strain (two items). A parallel examination of the factor components of the 12-item ZBI was performed. The model demonstrated a reasonable fit to the data based on the five preset fit criteria ($\chi 2$/df = 1.387; p = 0.550, CFI = 0.99, AGFI = 0.95, TLI = 0.99, and RMSEA = 0.036 (95% CI [0.023, 0.051]). All items in the model were significantly loaded on their respective factors (all p-values < 0.05), except for the factor-constraint item, for which no significance test could be conducted. In addition, all factor loadings were greater than 0.30, indicating that individual ZBI items were suitable for measuring the underlying construct of caregiver burden, albeit to varying degrees (**Fig 1**).

### Reliability analysis

The CR of the two domains of the 12-item ZBI ranged between 0.70 and 0.91, indicating acceptable reliability. With Cronbach's alpha values of 0.83 for the overall scale, 0.88 for personal strain, and 0.75 for role strain, the 12-item ZBI

**Table 1. Demographic characteristics of informal caregivers (n = 305).**

| Baseline characteristic | Number | Percent |
|---|---|---|
| Gender | | |
| Male | 63 | 20.6 |
| Female | 242 | 79.4 |
| Age (years) | | |
| Mean (SD) | 54.18 (13.52) | |
| Median (min: max) | 54 (30:86) | |
| Education level | | |
| Primary | 3 | 1.0 |
| Secondary | 218 | 71.4 |
| Bachelor's or higher | 84 | 27.6 |
| Religion | | |
| Buddhism | 255 | 83.9 |
| Islam | 49 | 16.1 |
| Occupation | | |
| Employed full-time | 193 | 63.3 |
| Employed part-time | 47 | 15.4 |
| Unemployed | 65 | 21.3 |
| Household income (baht) | | |
| Less than 5000 | 89 | 30.2 |
| 5,000-9,999 | 74 | 25.1 |
| 10,000-14,999 | 47 | 15.9 |
| 15,000-19,999 | 25 | 8.5 |
| ≥ 20,000 | 60 | 20.3 |
| Marital status | | |
| Single | 51 | 16.7 |
| Married | 208 | 68.2 |
| Divorced/ separated | 46 | 15.1 |
| Caregiver with illness | | |
| No | 171 | 56.1 |
| Yes | 134 | 43.9 |
| Relationship with patient | | |
| Parent | 34 | 11.2 |
| Spouse | 50 | 16.4 |
| Child | 149 | 48.8 |
| Other relatives (friends, neighbors) | 72 | 23.6 |
| Family members living with the patient | | |
| Less than 5 | 199 | 65.3 |
| ≥ 5 | 106 | 34.7 |
| Care hours per day | | |
| Between 1 and 5 | 30 | 10.0 |
| Between 6 and 10 | 36 | 12.0 |
| Between 11 and 15 | 51 | 17.0 |
| Between 16 and 20 | 19 | 6.3 |
| Between 21 and 24 | 164 | 54.7 |
| Mean (SD) | 17.72 (7.67) | |
| Median (min: max) | 24 (1:24) | |

*(Continued)*

**Table 1.** (Continued)

| Baseline characteristic | Number | Percent |
|---|---|---|
| Duration of caregiving (years) | | |
| Less than 5 | 227 | 74.4 |
| ≥ 5 | 78 | 25.6 |
| Diagnosis of patient | | |
| Cancer | 36 | 11.9 |
| Neurological disease | 100 | 33.1 |
| Cardiovascular disease | 17 | 5.6 |
| Disease of the elderly/Alzheimer's disease | 69 | 22.8 |
| Other | 85 | 26.6 |

**Table 2. Distribution of responses and discrimination of the items.**

| Item | Response (%) | | | | | Item-rest correlation |
|---|---|---|---|---|---|---|
| | 0 | 1 | 2 | 3 | 4 | |
| 1 | 36.7 | 50.0 | 0.0 | 10.0 | 3.3 | 0.49 |
| 2 | 41.3 | 46.0 | 0.0 | 10.0 | 2.7 | 0.70 |
| 3 | 66.7 | 32.7 | 0.0 | 0.7 | 0.0 | 0.82 |
| 4 | 77.3 | 20.7 | 0.0 | 1.3 | 0.7 | 0.83 |
| 5 | 40.7 | 46.7 | 0.0 | 10.0 | 2.7 | 0.80 |
| 6 | 60.0 | 30.0 | 0.0 | 8.7 | 1.3 | 0.80 |
| 7 | 59.3 | 35.3 | 0.0 | 4.7 | 0.7 | 0.81 |
| 8 | 70.7 | 25.3 | 0.0 | 1.3 | 2.7 | 0.81 |
| 9 | 64.0 | 28.7 | 0.0 | 6.0 | 1.3 | 0.80 |
| 10 | 60.7 | 35.3 | 0.0 | 3.3 | 0.7 | 0.81 |
| 11 | 36.0 | 42.7 | 0.0 | 13.3 | 8.0 | 0.84 |
| 12 | 30.7 | 44.0 | 0.0 | 10.0 | 15.3 | 0.85 |

demonstrated adequate internal consistency reliability. Similarly, this scale had strong internal consistency, as indicated by the item-rest correlations for all three subscales, which were greater than 0.30, and the corrected item-rest correlations, ranging from 0.38 to 0.70 (**Table 4**).

## Convergent validity

Table 4 contains details on the reliability analysis of the ZBI-12. AVE calculations revealed that all 12-item ZBI subscales had a convergence effect (with the AVE of personal strain = 0.50, and the AVE of role strain = 0.53). Furthermore, the square root of the ZBI's AVE (0.71) was higher than the correlation between the ZBI and DASS-21 (0.51). Based on the findings, we may conclude that the ZBI ratings have evidence of validity due to their relationship with other variables (convergent and discriminant evidence).

## Concurrent validity

External validity of the 12-item ZBI and the personal strain subscales is supported by correlation with concurrent measures. These scales showed a statistically significant positive and moderate to large association with the

**Table 3. Factor loading values for each item of the scale (n = 150).**

| ZBI item | Factor 1 | Factor 2 |
|---|---|---|
| | **Personal strain** | **Role strain** |
| 1. Don't have enough time for yourself | 0.590 | |
| 2. Stressed about fulfilling different responsibilities | 0.648 | |
| 3. Feel angry around the patient | 0.767 | |
| 4. Negative effect on other relationships | 0.678 | |
| 5. Feel strained around the patient | 0.770 | |
| 6. Health effects from caregiving | 0.752 | |
| 7. Have inadequate privacy | 0.808 | |
| 8. Suffering in social life | 0.772 | |
| 9. Sense of losing control over life | 0.831 | |
| 10. Feel uncertain of what to do | 0.648 | |
| 11. Feel you should be doing more for the patient | | 0.861 |
| 12. Feel things could be better for the patient | | 0.879 |
| KMO | 0.89 | |
| Cronbach's Alpha | 0.92 | 0.73 |
| Eigenvalue | 3.8 | 2.97 |
| % of variance | 46.83 | 14.91 |
| Cumulative % | 46.83 | 61.74 |

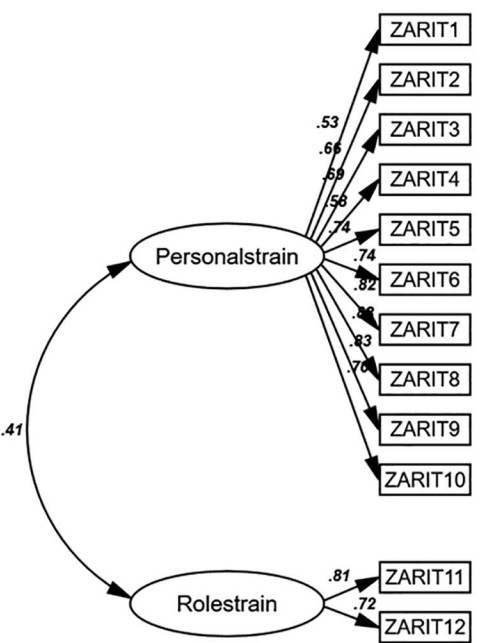

**Fig 1. Structure model of the 12-item ZBI with standardized path coefficient.**

DASS-depression, DASS-anxiety, and DASS-stress measures, with correlation ranging from 0.410.94. However, for the role strain, only the DASS-depression measure showed a statistically significant association with the role strain subscales, and the effect size was low (r = 0.17; p < 0.05) (**Table 5**).

 

**Table 4. Reliability analysis of the 12-item ZBI.**

| Subscale | Composite reliability | Cronbach's alpha | Item-total correlation | AVE |
|---|---|---|---|---|
| Personal strain | 0.91 | 0.88 | 0.38-0.70 | 0.50 |
| Role strain | 0.70 | 0.75 | 0.61-0.62 | 0.53 |
| Overall | | 0.83 | | |

## Discussion

The objective of this study was to validate the psychometric properties of the 12-item ZBI in Thai caregivers of palliative patients. The findings of this study show that the 12-item ZBI has acceptable psychometric qualities and has validity for use in palliative caregivers with high satisfaction and comprehension. The analysis of structural validity using EFA and CFA supported the two factors of the 12-item ZBI, which were as follows: personal strain (10 items), and role strain (two items) on the 12-item ZBI scale for this population. These two factors are based on the factor structure of the ZBI-22 described by Whitlatch [37]. Typically, "personal strain" refers to caregivers' subjective and emotional reactions to many aspects of caregiving. The term "role strain" refers to the practical aspects of caregiving, rather than an individual's personal experience. Indeed, it can be adequate. described as "the stress due to role conflict or overload" [37]. The results of this study were comparable with previous studies that validated this tool in caregivers of people diagnosed with dementia [13,16]. However, this result is also consistent with the psychometric study of the 12-item ZBI in Thailand among a psychiatric sample that reported a three-factor structure, which consisted of the impact, stress and care domains [15]. In addition, our factor analysis did not support the previously reported one-dimensionality of ZBI, which was verified in the sample of dementia caregivers [14]. The inconsistent results from factor analysis in different studies could be attributed to various components such as caring for patients with different conditions and needs varying levels of caregiver involvement. Furthermore, the nature of the relationships between patients and caregivers may contribute to understanding this occurrence. Regardless, the variation in factor dimensions among studies supports using only the ZBI total score rather than attempting to use subscale scores based on component dimensions [22].

The CR of the two domains of the 12-item ZBI ranged between 0.71 and 0.91, indicating acceptable reliability. Similarly, the Cronbach's alpha value from our study demonstrated that the 12-item ZBI total scores and those of its subscales had good internal consistency. This coefficient varied from fair to exceptional in past research that included both physiological and psychological caregivers [10,13–17,22], and the data revealed that the internal consistency of the 12-item ZBI was satisfactory and comparable to other studies [10,15–17,19]. Furthermore, the item analysis results showed that each scale's items had good discrimination indices. All item-total correlations were strongly and positively associated with the total score, demonstrating the scale's homogeneity. These indices suggested it might be possible to distinguish between high and low scores on this scale using the Thai-version items of the 12-item ZBI. Moreover, the tool's robust dependability suggests that it is skillfully crafted and capable of promptly addressing the query, as evidenced by the participants' feedback. This documentation verified the fact that the 12-item ZBI held significance and was well regarded by palliative caregivers in Thailand. Related research has also shown that this assessment instrument delivers an excellent item discrimination index [38,39].

The convergent validity of the 12-item ZBI was assessed by calculating the AVE for both subscales. The findings indicated that the AVE for all subscales was above 0.50, meeting the threshold for accepting convergent validity. These findings agree with the results of a study in China, which aimed to confirm the validity of the ZBI among caregivers of individuals with schizophrenia [38].

The content validity of the measures suggested that they had been specifically developed to assess higher levels of burden. However, caregivers reported experiencing lower levels of burden. This suggests that there may be a mismatch

**Table 5. Correlation between the 12-item ZBI and concurrent measures.**

| | 1 | 2 | 3 | 4 | 5 | 6 | 7 |
|---|---|---|---|---|---|---|---|
| 1. Total ZBI | 1 | | | | | | |
| 2. ZBI personal strain | 0.935** | 1 | | | | | |
| 3. ZBI role strain | 0.492** | 0.152 | 1 | | | | |
| 4. DASS-anxiety | 0.376** | 0.408** | 0.047 | 1 | | | |
| 5. DASS-depression | 0.476** | 0.470** | 0.172* | 0.763** | 1 | | |
| 6. DASS-stress | 0.562** | 0.575** | 0.156 | 0.799** | 0.820** | 1 | |
| 7. DASS-21 | 0.512** | 0.526** | 0.137 | 0.914** | 0.925** | 0.946** | 1 |

*p < .05 and

**p < 0.01 (two-tailed).

between the sample and the measure, leading to inaccurate negative burden ratings. Participation bias may account for the floor effects seen, as caregivers who elected to participate in the study may have had lower levels of stress compared to those who chose not to participate. Kühnel et al. reported a comparable account of caregiver participation bias [10]. In addition, the caregivers in this study were selected from two PC settings, which potentially decreased their burden more than that of caregivers who had limited access to professional assistance. These findings are consistent with results from prior studies among both advanced-cancer caregivers and palliative caregivers [10,40]. The findings of this study were comparable to those of a previous study that examined the challenges faced by caregivers of elderly Chinese individuals with dementia. The lower levels of burden observed in the previous study could potentially be attributed to the higher level of support received by caregivers as more than one-third of family caregivers in Hong Kong reported receiving help from domestic assistants [41].

Concurrent validity of the 12-item ZBI and the personal strain subscales is supported by correlation with DASS-21, especially the DASS-depression and DASS-anxiety measures. According to theory, caregivers often encounter a range of emotional and social difficulties. The results can be attributed to the fact that negative emotions were only linked to caregivers' depressive symptoms, suggesting that the adverse experience of caregiving had a greater impact on caregivers' mental well-being than the impairment and distress of care recipients. However, this does not imply that other dimensions of caregiver burden are nonexistent. This highlights the necessity for further research on other factors that may contribute to increasing stress among caregivers. This data is consistent with two previous research studies, which found a significant association between caregiving burden and depressive symptoms in caregivers of heart failure patients [20]. Another study reported significant positive correlations among the 12-item ZBI, the Beck Depression Inventory, and the Beck Anxiety Inventory in caregivers of cancer patients [39]. This conclusion demonstrates that ZBI is a valid indicator of negative emotional states in caregivers of palliative patients as its validity has already been established in caregivers of patients with other chronic illnesses. Conversely, ZBI's AVE was 0.50, which was sufficient for the specific setting of the investigation, and the markers of discriminant evidence were within acceptable thresholds. Therefore, ZBI scores offer both convergent and discriminant evidence.

## Strengths and limitations

The key component of this study is that it is the first to investigate the structure and psychometric properties of the 12-item ZBI among palliative caregivers in Thailand. This research is subject to certain constraints, including a few methodological problems. Initially, we examined palliative caregivers from two district health service networks in the southern area of Thailand. However, it is important to note that these caregivers may not have been representative of the entire population of palliative caregivers. Moreover, this study employed a methodological study in which data was collected at only one point. It does not allow for the evaluation of test-retest reliability or long-term measurement stability. As a

result, it limits the capacity to draw conclusions regarding the 12-item ZBI's long-term usefulness in measuring caregiver burden.

## Conclusion

This study offers psychometric validation for the 12-item ZBI as a reliable tool to assess the burden of care experienced by palliative caregivers. We have gathered compelling evidence to support reliability, convergent validity, and construct validity in this particular population. Factor analysis confirms a two-dimensional internal structure. As a result, this version of the ZBI enables a more efficient and useful burden assessment and health care providers can utilize this tool in both research and clinical environments to evaluate the level of burden experienced by palliative caregivers. Furthermore, the additional validation studies of the 12-item ZBI using longitudinal design are warranted to ensure its long-term measurement stability.

## Supporting information

**S1 Data. Zarit data.**
(XLSX)

## Acknowledgments

The authors would like to thank Dr. Orawan Silpakit for granting permission to use the Thai version of the 12-item ZBI, as well as all the caregivers who participated in this study. We value the participants' readiness to furnish their personal information for the purpose of this research.

## Author contributions

**Conceptualization:** Ueamporn Summart, Thanaporn Kaewbubpha, Ornicha Saengnil, Prasertsri Kongkaur, Suparuek Mahorasop, Suchittra Saengkhum, Yuwadee Wittayapun.

**Data curation:** Yuwadee Wittayapun.

**Formal analysis:** Ueamporn Summart, Yuwadee Wittayapun.

**Investigation:** Thanaporn Kaewbubpha, Ornicha Saengnil, Prasertsri Kongkaur, Suparuek Mahorasop, Suchittra Saengkhum, Yuwadee Wittayapun.

**Methodology:** Ueamporn Summart, Thanaporn Kaewbubpha, Ornicha Saengnil, Prasertsri Kongkaur, Suparuek Mahorasop, Suchittra Saengkhum, Yuwadee Wittayapun.

**Project administration:** Thanaporn Kaewbubpha, Ornicha Saengnil, Prasertsri Kongkaur, Suparuek Mahorasop, Suchittra Saengkhum, Yuwadee Wittayapun.

**Resources:** Yuwadee Wittayapun.

**Software:** Ueamporn Summart, Yuwadee Wittayapun.

**Supervision:** Yuwadee Wittayapun.

**Validation:** Ueamporn Summart, Thanaporn Kaewbubpha, Ornicha Saengnil, Prasertsri Kongkaur, Suparuek Mahorasop, Suchittra Saengkhum, Yuwadee Wittayapun.

**Visualization:** Ueamporn Summart, Yuwadee Wittayapun.

**Writing – original draft:** Ueamporn Summart, Yuwadee Wittayapun.

**Writing – review & editing:** Ueamporn Summart, Thanaporn Kaewbubpha, Ornicha Saengnil, Prasertsri Kongkaur, Suparuek Mahorasop, Suchittra Saengkhum, Yuwadee Wittayapun.

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
