## [Decision Letter · Decision Letter 0]

24 Feb 2025

PONE-D-24-26645Validation of the Thai version of the 12-item Zarit Burden Interview among informal caregivers in a palliative care settingPLOS ONE

Dear Dr. Wittayapun,

Thank you for submitting your manuscript to PLOS ONE. After careful consideration, we feel that it has merit but does not fully meet PLOS ONE’s publication criteria as it currently stands. Therefore, we invite you to submit a revised version of the manuscript that addresses the points raised during the review process.

We look forward to receiving your revised manuscript.

Kind regards,

Madson Alan Maximiano-Barreto, Ph.D

Academic Editor

PLOS ONE

Journal Requirements:

2. Please amend your list of authors on the manuscript to ensure that each author is linked to an affiliation. Authors’ affiliations should reflect the institution where the work was done (if authors moved subsequently, you can also list the new affiliation stating “current affiliation:….” as necessary).

Reviewers' comments:

Reviewer's Responses to Questions

**Comments to the Author**

1. Is the manuscript technically sound, and do the data support the conclusions?

Reviewer #1: Yes

Reviewer #2: Yes

2. Has the statistical analysis been performed appropriately and rigorously? 

Reviewer #1: Yes

Reviewer #2: Yes

3. Have the authors made all data underlying the findings in their manuscript fully available?

Reviewer #1: No

Reviewer #2: Yes

4. Is the manuscript presented in an intelligible fashion and written in standard English?

Reviewer #1: Yes

Reviewer #2: Yes

5. Review Comments to the Author

Reviewer #1: Methods:

- Lines 108-112: If the authors included friends as caregivers, they should avoid calling them “family caregivers”. Informal (non-paid) caregivers seem a better term if friends are also included.

- Lines 112-113: Do the authors know the minimum hours per day that were provided each day?

- Lines 122-123: Is the 12-item Zarit Burden Interview scale validated in Thai? Or just the 22-item version? This is very important. If it is not validated, it should be mentioned in the limitations section.

- Lines 138-140: Please detail the randomization process. How was it performed?

- Line 144: Please change to “All statistical analyses in this study were conducted using…”

Results:

- I have several comments for Table 1:

o Please specify the “other relative”. How many of those were friends or non-family caregivers?

o What do you mean by caregiver with illness? What type of illness? There is a difference between having dyslipidemia or having rheumatoid arthritis, for example.

o Family members living with the caregiver or with the patient?

o Providing care between 1 and 10 hours per week (22%) seems too little to me. If one of the inclusion criteria includes providing care at least three days a week, how is it possible to provide 1 hour of care? This is a bit confusing. I can’t believe someone provides only 20 minutes per care a day.

- Convergent and concurrent validity: The baseline characteristics of the population are extremely heterogeneous. It is not the same to provide care 1 hour per week or 24 hours. Same point for family vs. non-family caregivers. I think the authors should perform sub-analysis to identify the impact of baseline characteristics in these results.

Limitations:

- The limitations section would benefit from including all the actual limitations of the study. Most of your caregivers have been providing care for a short time (<5 years) and for a very short number of hours per week (<10 hours, 22%). Are you using a 12-item Thai validated ZBI? Or just the 22-item ZBI has been validated?

Reviewer #2: This validation study of the ZBI-12 can contribute to clinical care, both in outpatient and hospital settings, as well as to scientific research, which can benefit from this instrument. I suggest that the authors review the English language.

6. PLOS authors have the option to publish the peer review history of their article (what does this mean? ). If published, this will include your full peer review and any attached files.

**Do you want your identity to be public for this peer review?** For information about this choice, including consent withdrawal, please see our Privacy Policy .

Reviewer #1: No

Reviewer #2: No

---

## [Author Response · Author response to Decision Letter 0]

4 Mar 2025

Responses to reviewers on MS version dated 3 March 2025

Psychometric validation of the Thai version of the 12-item Zarit Burden Interview among informal caregivers in a palliative care setting

Journal Requirements

Response:

Thank you for your suggestion for improving our manuscript. We have reviewed the PLOS ONE's style requirements and ensured that our manuscript complies with the specified formatting, including file naming conventions.

Please see the revised manuscript.

2) Please amend your list of authors on the manuscript to ensure that each author is linked to an affiliation. Authors’ affiliations should reflect the institution where the work was done (if authors moved subsequently, you can also list the new affiliation stating “current affiliation:….” as necessary).

Response:

Thank you for your comment regarding the authors' affiliations. We have ensured that the listed affiliations accurately reflect the institutions where the research was conducted. Also, our research team consists of members from both academic institutions and healthcare service settings, in alignment with our concept of the academic-practice approach. This interdisciplinary collaboration was essential in strengthening our research team.

Response to Editor:

1) Title: I suggest that the authors modify the title to "Psychometric Validation of the Thai Version of the 12-Item Zarit Burden Interview Among Informal [...]" to improve clarity and precision.

Response:

Thank you for your suggestion for improving our manuscript. We modified the title of our manuscript and followed your suggestion.

Please see the revised manuscript.

2) Introduction:

2.1) I suggest that the acronym “WHO” be written out in full the first time it appears, with WHO in parentheses.

Response:

Thank you for your suggestion. We added the acronym “WHO” in our manuscript.

(in line 49)

Please see the revised manuscript.

2.2) In the phrase “According to this description...”, I recommend that the authors include at least one reference to support this statement.

Response:

Thank you for your suggestion. We include one reference to support this statement following your suggestion. (in line 53)

Radbruch L, De Lima L, Knaul F, Wenk R, Ali Z, Bhatnaghar S, et al. Redefining palliative care—a new consensus-based definition. Journal of pain and symptom management. 2020;60(4):754-64. https:// doi: 10.1016/j.jpainsymman.2020.04.027. PMID: 32387576

Please see the revised manuscript

2.3) If the Zarit Burden Interview (ZBI) is one of the most widely used self-report questionnaires, I suggest including additional references to substantiate this claim

Response:

Thank you for your suggestion. As the Zarit Burden Interview (ZBI) is one of the most widely used self-report questionnaires, we include additional references to substantiate this claim. (in line 76)

Please see the revised manuscript

2.4) I also recommend that the justification for this study be more detailed. Since self-report questionnaires offer several advantages in clinical settings—such as low implementation costs and shorter administration times—highlighting these aspects could strengthen the rationale for the study.

Response:

Thank you for your suggestion. We added this statement to our manuscript to support several advantages of self-report questionnaires.

These self-report questionnaires provide various advantages in clinical settings, including minimal implementation costs and shorter administration times. (in line 78-79)

Please see the revised manuscript

2.5) Include justifications in the text for why informal caregivers were chosen as the sample.

Response:

Thank you for your suggestion. We included the statement of informal caregiver to clarify its definition and support our justification.

There are 2 types of caregivers formal and informal. The formal caregiver is the individual who is compensated for the service. The informal caregiver is typically a family member, friend, or neighbor. Although this type of caregiver is neither paid nor trained to conduct the function, they have a great degree of dedication as evidenced by their affective relationship [4]. In the case of palliative patients, the type of care in healthcare is usually informal, with the family member being the primary caregiver. This situation has impacted numerous caregivers of individuals with advanced or terminal illnesses, leading to a significant burden [5]. (in line 57-63)

Please see the revised manuscript

2.6) In line 131, I suggest that the authors specify the good psychometric properties of the DASS-21 Thailand and explain why this scale was chosen over another.

Response:

Thank you for your suggestion. We add the reliability of the Thai version of DASS-21 to support our justification.

The Cronbach’s alpha coefficient for the Thai version of the DASS-21 was 0.92.

(in line 141-142)

Please see the revised manuscript

3) Methods:

Study Design and Population:

I suggest that the inclusion and exclusion criteria be explicitly stated to ensure clarity

Response:

Thank you for your suggestion. We added the inclusion and exclusion criteria for clarifying the enrolled sample in this study.

The inclusion criteria for this study were the primary caregivers who included friends, family members, and individuals involved in PC, through patients receiving treatment in medical institutions or at home through two district health service networks. They were male or female Thai nationals, aged 18 or above, who had provided care for a minimum of 3 days per week for at least 3 months without payment [24]. The exclusion criteria were caregivers with poor general conditions or having mental disorders such as schizophrenia, psychosis, or dementia. (in line 117-122)

Please see the revised manuscript

3.1) What theoretical background was used to determine the criteria of three months and three days per week? I recommend providing a reference to support this choice.

Response:

Thank you for your suggestion. However, the duration of care varies across previous studies, depending on the condition of the patients. We attempted to provide a reference to support this statement (see line 120).

Koopman E, Heemskerk M, van der Beek AJ, Coenen P. Factors associated with caregiver burden among adult (19–64 years) informal caregivers–An analysis from Dutch Municipal Health Service data. Health & social care in the community. 2020;28(5):1578-89.

https://doi: 10.1111/hsc.12982. PMID: 32207221

Please see the revised manuscript

3.2) Since the sample includes both sexes, I suggest explicitly mentioning this in the inclusion criteria.

Response:

Thank you for your suggestion. We added this statement to our inclusion criteria.

They were male or female Thai nationals, aged 18 or above, who had provided care for a minimum of 3 days per week for at least 3 months without payment.

(in line 119-120)

Please see the revised manuscript

3.3) How were the participants randomly selected? This process needs to be described in detail

Response:

Thank you for your comments. The participants enrolled in this study did not randomly select as they came from the health service network that we mentioned in Study design and population. However, we included additional details to clarify this statement.

The family caregivers of palliative patients served by the hospital support team and the home care team were recruited consecutively. (in line 115-116)

Please see the revised manuscript

3.4) In line 131, I suggest that the authors specify the good psychometric properties of the DASS-21 Thailand and explain why this scale was chosen over another.

Response:

Thank you for your suggestion. We add the reliability of the Thai version of DASS-21 to support our justification.

The Cronbach’s alpha coefficient for the Thai version of the DASS-21 was 0.92.

(in line 141-142)

Please see the revised manuscript

4) Strengths and Limitations:

The results are well described and effectively represented in tables and figures. I suggest that the authors check the quality of Figure 1, as it appears blurry

Response:

Thank you for your suggestion. We edit our figure following the journal recommendations. (in line 241-243)

Please see the revised manuscript

5) Strengths and Limitations:

The limitations presented by the authors are valid and encourage reflections on how future studies can build upon these findings.

Response:

Thank you for your opinion. We appreciated your comments and attempt to improve our manuscript following your suggestion.

6) Conclusion:

• I suggest reformulating the conclusion, as some sentences included in it would be

more appropriate in the limitations section.

Response:

Thank you for your suggestion. We reformulate some parts of our conclusion that would be the imitation.

This study offers psychometric validation for the 12-item ZBI as a reliable tool to assess the burden of care experienced by palliative caregivers. We have gathered compelling evidence to support reliability, convergent validity, and construct validity in this population. Factor analysis confirms a two-dimensional internal structure. As a result, this version of the ZBI enables a more efficient and useful burden assessment and health care providers can utilize this tool in both research and clinical environments to evaluate the level of burden experienced by palliative caregivers.

(in line 355-361)

Please see the revised manuscript

7) I suggest that the authors review the manuscript's English.

Response:

Thank you for your suggestion. We reviewed the grammatical of our manuscript and he accuracy of English language, grammar, punctuation, spelling and overall style were edit and proof by native from International College for Sustainability Studies. We also sent the file of certification of language proof via the first submission.

Response to reviewer 1

1) Methods:

- Lines 108-112: If the authors included friends as caregivers, they should avoid calling them “family caregivers”. Informal (non-paid) caregivers seem like a better term if friends are also included.

Response:

Thank you for your suggestion for improving our manuscript. After we reviewed the definition of informal caregivers following your suggestion we agree with your comments and edited this term to “informal caregivers”

(In line 115)

Please see the revised manuscript.

1.1) Lines 112-113: Do the authors know the minimum hours per day that were provided each day?

Response:

Thank you for your comments. Please accept our apology for our error, however we reviewed and revised these minimum hours of caring per day and added the reference for clarifying this definition.

They were male or female Thai nationals, aged 18 or above, who had provided care for a minimum of 3 days per week for at least 3 months without payment [24].

(In line 119-121)

Please see the revised manuscript.

1.2) Lines 122-123: Is the 12-item Zarit Burden Interview scale validated in Thai? Or just the 22-item version? This is very important. If it is not validated, it should be mentioned in the limitations section.

Response:

Thank you for your comments. The Thai version of 12-item ZBI has been validated in caregiver of psychiatric patients.

The Thai version of the 12-item ZBI (ZBI-12) has been validated in of psychiatric patients with an excellent internal consistency [12].

(In line 128-129)

Please see the revised manuscript.

1.3) Lines 138-140: Please detail the randomization process. How was it performed?

Response:

Thank you for your comments. The participants enrolled in this study did not randomly select as they came from the health service network that we mentioned in Study design and population. However, we included additional details to clarify this statement.

The family caregivers of palliative patients served by the hospital support team and the home care team were recruited consecutively. (in line 115-116)

Please see the revised manuscript

1.4) Line 144: Please change to “All statistical analyses in this study were conducted using…”

Response:

Thank you for your suggestion for improving our manuscript.

We edited this statement following your suggestion. (in line 154)

Please see the revised manuscript

2) Results:

2.1) I have several comments for Table 1:

o Please specify the “other relative”. How many of those were friends or non-family caregivers?

Response:

Thank you for your suggestion for improving our results.

We added the details of other relatives (friends, neighbors) in table 1

Please see the revised manuscript

2.2) What do you mean by caregiver with illness? What type of illness? There is a difference between having dyslipidemia or having rheumatoid arthritis, for example.

Response:

Thank you for your comments about improving our results.

Please accept our apology as we did not collect the details of caregiver health problems in our questionnaire. The question for information details is whether the caregivers have any health problems or not. Thanks for your suggestion.

2.3) Family members living with the caregiver or with the patient?

Response:

Thank you for your comments about improving our results.

We added this detail with “Family members living with the patient”

Please see the revised manuscript

2.4) Providing care between 1 and 10 hours per week (22%) seems too little to me. If one of the inclusion criteria includes providing care at least three days a week, how is it possible to provide 1 hour of care? This is a bit confusing. I can’t believe someone provides only 20 minutes per care a day.

Response:

Thank you for your comments. Please accept our apology for our error as we collected data for both the number of day care /week and care hour/day and confused for using data analyses in our previous manuscript as you think someone provides only 20 minutes per care a day. (We are confused about the number of day care and number of hour care/day). However, we reviewed and revised this minimum care hours/ day instead of the wording existing in our previous day of care hour/week, and also added the reference for clarifying this definition.

They were male or female Thai nationals, aged 18 or above, who had provided care for a minimum of 3 days per week for at least 3 months without payment [24].

(In line 119-121). In addition, we reviewed and revised our analysis and edited the number of hours followed by both the definition and the number of care hour/day instead of the number care hour/week that existing in the previous manuscript.

Please see the revised manuscript

3) The limitations section would benefit from including all the actual limitations of the study. Most of your caregivers have been providing care for a short time (<5 years) and for a very short number of hours per week (<10 hours, 22%). Are you using a 12-item Thai validated ZBI? Or just the 22-item ZBI has been validated?

Response:

Thank you for your suggestion. After we reviewed our data and revised our analysis, we found some mistakes and accepted your comments. We carefully checked and edited these statements to clarify these results. We edited care hour/week to care hour/day following our data analysis.

Please see the revised manuscript

Response to reviewer 2

1) Limitations:

This validation study of the ZBI-12 can contribute to clinical care, both in outpatient and hospital settings, as well as to scientific research, which can benefit from this instrument. I suggest that the authors review the English language.

Response 9:

Thank you for your suggestion.

We edited these statements following your suggestion in the conclusion of this study.

(In line 358-362).

Please see the revised manuscript

I would like to express my gratitude for your diligent efforts in providing comments and suggestions to improve our manuscript. We sincerely appreciate your feedback and have revised the manuscript accordingly. If you have any further comments, we are happy to make additional edits and resubmit the manuscript as quickly as possible.

We have corrected the data in Table 1, as we initially confused the number of care

---

## [Decision Letter · Decision Letter 1]

14 Mar 2025

PONE-D-24-26645R1Psychometric validation of the Thai version of the 12-item Zarit Burden Interview among informal caregivers in a palliative care settingPLOS ONE

Dear Dr. Wittayapun,

Thank you for submitting your manuscript to PLOS ONE. After careful consideration, we feel that it has merit but does not fully meet PLOS ONE’s publication criteria as it currently stands. Therefore, we invite you to submit a revised version of the manuscript that addresses the points raised during the review process.

We look forward to receiving your revised manuscript.

Kind regards,

Madson Alan Maximiano-Barreto, Ph.D

Academic Editor

PLOS ONE

**Journal Requirements:**

Reviewers' comments:

Reviewer's Responses to Questions

**Comments to the Author**

1. If the authors have adequately addressed your comments raised in a previous round of review and you feel that this manuscript is now acceptable for publication, you may indicate that here to bypass the “Comments to the Author” section, enter your conflict of interest statement in the “Confidential to Editor” section, and submit your "Accept" recommendation.

Reviewer #2: All comments have been addressed

Reviewer #3: (No Response)

2. Is the manuscript technically sound, and do the data support the conclusions?

Reviewer #2: Yes

Reviewer #3: Yes

3. Has the statistical analysis been performed appropriately and rigorously? 

Reviewer #2: Yes

Reviewer #3: Yes

4. Have the authors made all data underlying the findings in their manuscript fully available?

Reviewer #2: Yes

Reviewer #3: Yes

5. Is the manuscript presented in an intelligible fashion and written in standard English?

Reviewer #2: No

Reviewer #3: Yes

6. Review Comments to the Author

**Reviewer #2: ** Dear author,

I would like to thank you for making all the necessary changes and accepting the suggestions in this report. I congratulate the authors on their excellent work, which will certainly add value to the scientific literature. I hope this manuscript will be available to the community soon.

**Reviewer #3: ** Dear Authors,

Congratulations on choosing this important topic. I believe that validating this scale ensures that the tool is culturally appropriate and reliable for measuring caregiver burden in Thailand, which is crucial for accurate future assessments and interventions. This can lead to better support and resources for caregivers, ultimately improving their well-being and the quality of care provided to patients.

However, for the publication of the manuscript, I suggest the following comments and questions, for example:

1) Regarding the study’s limitations, it is necessary to consider:

- The study employed a cross-sectional design, meaning that data were collected at a single point in time. This design does not allow for the assessment of test-retest reliability or the stability of measurements over time. Consequently, it limits the ability to draw conclusions about the long-term applicability of the 12-item Zarit Burden Interview (ZBI) in measuring caregiver burden.

- The confirmatory factor analysis indicated that negative emotions were primarily linked to caregivers’ depressive symptoms. This suggests that the adverse experiences of caregiving may have a more significant impact on mental well-being than the distress experienced by care recipients. This focus may overlook other important dimensions of caregiver burden.

2) Other comments:

- The study is better classified as a methodological study, although some publications refer to it as a “cross-sectional study.” I suggest reviewing this classification based on the available literature.

- There are minor grammar and typographical errors, such as "circumstance" in the abstract, which should be "context." I suggest reviewing the grammar and spelling of the text.

- The justification for the division into "personal strain" and "role strain" could be better supported theoretically.

- The layout of the tables should be revised, or they should be labeled as "figures." Additionally, the titles should include the period and location (country) where the study was conducted.

- The conclusion should be revised in light of the considerations/comments presented.

7. PLOS authors have the option to publish the peer review history of their article (what does this mean? ). If published, this will include your full peer review and any attached files.

**Do you want your identity to be public for this peer review?** For information about this choice, including consent withdrawal, please see our Privacy Policy .

Reviewer #2: No

Reviewer #3: No

---

## [Author Response · Author response to Decision Letter 1]

20 Mar 2025

Responses to reviewers on MS version dated 20 March 2025

Psychometric validation of the Thai version of the 12-item Zarit Burden Interview among informal caregivers in a palliative care setting in the south of Thailand.

Journal Requirements

1) Please review your reference list to ensure that it is complete and correct. If you have cited papers that have been retracted, please include the rationale for doing so in the manuscript text or remove these references and replace them with relevant current references. Any changes to the reference list should be mentioned in the rebuttal letter that accompanies your revised manuscript. If you need to cite a retracted article, indicate the article’s retracted status in the References list and also include a citation and full reference for the retraction notice.

Response:

Thank you for your suggestion for improving our manuscript. We reviewed all references in this manuscript and no papers in our manuscript that have been retracted.

In addition, we added 2 new references following the reviewer’s suggestion in reference order no. 24 and order no. 37, so the reference list has been changed as follows:

24. Mbuagbaw L, Lawson DO, Puljak L, Allison DB, Thabane L. A tutorial on methodological studies: the what, when, how and why. BMC Medical Research Methodology. 2020;20:1-12.

https://doi.org/10.1186/s12874-020-01107-7. PMID: 32894052.

37. Whitlatch CJ, Zarit SH, von Eye A. Efficacy of interventions with caregivers: a reanalysis. The

Gerontologist. 1991;31(1):9-14. https://doi.org/ 10.1093/geront/31.1.9. PMID: 2007480.

Please see the revised manuscript.

Response to Reviewer 2:

1) I would like to thank you for making all the necessary changes and accepting the suggestions in this report. I congratulate the authors on their excellent work, which will certainly add value to scientific literature. I hope this manuscript will be available to the community soon

Response:

Thank you for all your suggestions to help us improve our manuscript.

Response to reviewer 3

1) Regarding the study’s limitations, it is necessary to consider:

1.1) The study employed a cross-sectional design, meaning that data were collected at a single point in time. This design does not allow for the assessment of test-retest reliability or the stability of measurements over time. Consequently, it limits the ability to draw conclusions about the long-term applicability of the 12-item Zarit Burden Interview (ZBI) in measuring caregiver burden.

Response:

Thank you for your suggestion for improving our manuscript. We agreed with your opinion and added these statements in both part of the limitation and the conclusion of our study.

Limitation:

Moreover, this study employed a methodological study in which data was collected at a single point in time. It does not allow for the evaluation of test-retest reliability or long-term measurement stability. As a result, it limits the capacity to draw conclusions regarding the 12-item ZBI's long-term usefulness in measuring caregiver burden. (In line 356-359)

Conclusions:

Furthermore, the additional validation studies of the 12-item ZBI using longitudinal design are warranted to ensure its long-term measurement stability.

(In line 368-369)

Please see the revised manuscript.

1.2) The confirmatory factor analysis indicated that negative emotions were primarily linked to caregivers’ depressive symptoms. This suggests that the adverse experiences of caregiving may have a more significant impact on mental well-being than the distress experienced by care recipients. This focus may overlook other important dimensions of caregiver burden.

Response:

Thank you for your comments and suggestions. After we reviewed previous studies, our results are comparable to other studies that aimed to validate the 12-item Zarit Burden Interview; however, some studies reported another dimension, such as "guilt." It remains possible that a three-factor structure is unique to the 22-item ZBI, but it accounts for the lack of a third factor found with the 12-item ZBI in this study and previous studies. As a result, please allow us to add your additional details to our discussion part.

However, this does not imply that other dimensions of caregiver burden are nonexistent. This highlights the necessity for further research on other factors that may contribute to increasing stress among caregivers. (In line 337-340)

Please see the revised manuscript.

2) Other comments:

2.1) The study is better classified as a methodological study, although some publications refer to it as a “cross-sectional study.” I suggest reviewing this classification based on the available literature.

Response:

Thank you for your suggestion. We reviewed and edited our study design, followed your suggestion, and added the reference to support this classification.

This methodological study focused on the validity of the 12-item ZBI, utilizing data from the "palliative care outcomes and factors associated with the quality of life of caregivers of palliative care patients" study [24]., (In line 109-111)

24. Mbuagbaw L, Lawson DO, Puljak L, Allison DB, Thabane L. A tutorial on methodological studies: the what, when, how and why. BMC Medical Research Methodology. 2020; 20:1-12.

https://doi.org/10.1186/s12874-020-01107-7. PMID: 32894052.

Please see the revised manuscript

2.2) There are minor grammar and typographical errors, such as "circumstance" in the abstract, which should be "context." I suggest reviewing the grammar and spelling of the text.

Response:

Thank you for your suggestion for improving our manuscript.

We edited this statement following your suggestion, and we carefully reviewed the grammar and spelling of the text in this manuscript.

(In line 24)

Please see the revised manuscript

2.3) The justification for the division into "personal strain" and "role strain" could be better supported theoretically.

Response:

Thank you for your suggestion for improving our results.

We added the details and definitions of the 2 factors, namely "personal strain" and "role strain," that existed in the original version of the 12-item ZBI.

These two factors are based on the factor structure of the ZBI-22 described by Whitlatch [37]. Typically, "personal strain" refers to caregivers' subjective and emotional reactions to many aspects of caregiving. The term "role strain" refers to the practical aspects of caregiving, rather than an individual's personal experience. Indeed, it can be adequate. described as "the stress due to role conflict or overload" [37].

(In line 281-285)

37. Whitlatch CJ, Zarit SH, von Eye A. Efficacy of interventions with caregivers: a reanalysis. The

Gerontologist. 1991;31(1):9-14. https://doi.org/ 10.1093/geront/31.1.9. PMID: 2007480.

Please see the revised manuscript

2.4) The layout of the tables should be revised, or they should be labeled as "figures." Additionally, the titles should include the period and location (country) where the study was conducted.

Response:

Thank you for your comments about improving our results.

We present our CFA with figure 1. Regarding the layout of the tables, after careful consideration, we have decided to maintain their current format, as they effectively convey the data in a clear and concise manner.

We included the setting of our study in the titles.

Psychometric validation of the Thai version of the 12-item Zarit Burden Interview among informal caregivers in a palliative care setting in the south of Thailand.

Please see the revised manuscript

2.5) The conclusion should be revised in light of the considerations/comments presented.

Response:

Thank you for your comments about improving our conclusion.

Furthermore, the additional validation studies of the 12-item ZBI using longitudinal design are warranted to ensure its long-term measurement stability.

(In line 368-369)

Please see the revised manuscript

Finally, we would like to express our sincere gratitude to the reviewers for their thoughtful and constructive feedback on our manuscript. Their insightful comments have been invaluable in refining our work and enhancing its clarity and quality. We appreciate any additional feedback you may have and are ready to make further revisions. We will resubmit the manuscript promptly after addressing your comments.

---

## [Decision Letter · Decision Letter 2]

30 Mar 2025

Psychometric validation of the Thai version of the 12-item Zarit Burden Interview among informal caregivers in a palliative care setting in the south of Thailand.

PONE-D-24-26645R2

Dear Dr. Wittayapun,

We’re pleased to inform you that your manuscript has been judged scientifically suitable for publication and will be formally accepted for publication once it meets all outstanding technical requirements.

Kind regards,

Madson Alan Maximiano-Barreto, Ph.D

Academic Editor

PLOS ONE

Additional Editor Comments (optional):

Dear authors, 

I am happy to see your efforts to meet the suggestions, as well as respond point by point to the reviewers' comments. After carefully evaluating your manuscript, I see that it is suitable to be accepted and thus published in Plos One.

---

## [Editor Report · Acceptance letter]

PONE-D-24-26645R2

PLOS ONE

Dear Dr. Wittayapun,

I'm pleased to inform you that your manuscript has been deemed suitable for publication in PLOS ONE. Congratulations! Your manuscript is now being handed over to our production team.

Kind regards,

on behalf of

Dr. Madson Alan Maximiano-Barreto

Academic Editor

PLOS ONE